



# Deconvolution of Boundary Layer Depth and Aerosol Constraints on Cloud Water Path in Subtropical Stratocumuli

Anna Possner[1], Ryan Eastman[2], Frida Bender[3], and Franziska Glassmeier[4]

[1]Institute for Atmospheric and Environmental Sciences, Goethe University,Frankfurt/Main, Germany
[2]Department of Atmospheric Sciences, University of Washington, Seattle, USA
[3]Department of Meteorology and Bolin Centre for Climate Research, Stockholm University, Stockholm, Sweden
[4]Department of Environmental Sciences,Wageningen University, Wageningen, Netherlands

**Correspondence:** Anna Possner (apossner@iau.uni-frankfurt.de)

**Abstract.** The liquid water path ($LWP$) adjustment due to aerosol-cloud interactions in marine stratocumuli remains a considerable source of uncertainty for climate sensitivity estimates. An unequivocal attribution of $LWP$ changes to changes in aerosol concentration from climatology remains difficult due to the considerable covariance between meteorological conditions alongside changes in aerosol concentrations. Here, we show that $LWP$ susceptibility in marine boundary layers (BLs) inferred from climatological relationships, triples in magnitude from $-0.1$ to $-0.33$ as the BL deepens.

We further find deep BLs to be underrepresented in pollution track, process modelling and in-situ studies of aerosol-cloud interactions in marine stratocumuli. Susceptibility estimates based on these approaches are skewed towards shallow BLs of moderate $LWP$ susceptibility. Therefore, extrapolating $LWP$ susceptibility estimates from shallow BLs to the entire cloud climatology, may underestimate the true $LWP$ adjustment within sub-tropical stratocumuli, and thus overestimate the effective aerosol radiative forcing in this region.

Meanwhile, $LWP$ susceptibility estimates inferred from climatology in deep BLs are still poorly constrained. While susceptibility estimates in shallow BLs are found to be consistent with process modelling studies, they are overestimated as compared to pollution track estimates.

## 1 Introduction

The aerosol radiative forcing due to by changes in cloud reflectivity of low-level marine clouds remains one of the largest sources of physical uncertainty in climate sensitivity estimates. Estimates in total aerosol radiative forcing from the Fifth Assessment Report (AR5) issued by the Intergovernmental Panel on Climate Change (IPCC) range from $-0.1\,\mathrm{W\,m^{-2}}$ to $-1.9\,\mathrm{W\,m^{-2}}$ (Boucher et al., 2013; Zelinka et al., 2014). Based on these estimates, increased cloud reflectivity due to anthropogenic aerosol, may have posed a substantial offset to the greenhouse gas forcing.

However, this cooling term is likely to reduce in coming years as anthropogenic emissions of aerosols decline (Smith and Bond, 2014). Yet, the quantification of aerosol induced changes in cloud scene albedo remains important for reducing the uncertainty in overall forcing. Subtropical marine stratocumuli are of particular relevance; the stratocumulus decks in the subtropics contribute strongly to the cooling of the planet by reflecting $\sim 40\%$ of incoming solar radiation on average, in a region of high



solar intensity (Bender et al., 2011).

In particular, cloud adjustments to changes in aerosol concentration remain highly uncertain. As defined in IPCC AR5 (Boucher et al., 2013), adjustments quantify the net response in cloud-radiative properties to external forcing agents such as anthropogenic aerosols. Through dynamic or thermodynamic adjustments, such as decreased precipitation rates (Albrecht, 1989), increased mixing rates at cloud top (Ackerman et al., 2004), or the sedimentation-entrainment feedback (Bretherton et al., 2007), the thermodynamics of the cloud is impacted and the liquid water path ($LWP$) may be altered. Adjustments in cloud

fraction ($CF$) by changes in aerosol concentration may also increase the overall albedo of the cloud scene (Gryspeerdt et al., 2016; Andersen et al., 2017; Possner et al., 2018). However, these effects cannot be addressed within the framework of this study due to the insufficient accuracy in $CF$ retrievals under polluted conditions (e.g. Twohy et al. (2009)). It is therefore mentioned here for completeness, but will not be discussed further.

In order to constrain the uncertainty range reflected within the wide range of AR5 forcing estimates, numerous studies have

since quantified the individual contributions of the Twomey effect (Twomey, 1991) and $LWP$ adjustments in global-scale and long-term satellite records (Sekiguchi et al., 2003; Quaas et al., 2008; Lebsock et al., 2008; Bellouin et al., 2013; Bender et al., 2016; Gryspeerdt et al., 2017; McCoy et al., 2017; Gryspeerdt et al., 2019; Rosenfeld et al., 2019), pollution track data sets (Ackerman et al., 2000; Christensen and Stephens, 2011; Christensen et al., 2014; Chen et al., 2015; Malavelle et al., 2017; Toll et al., 2017; Bender et al., 2019; Toll et al., 2019), and large-eddy (LES) or cloud-resolving simulations in com-

bination with field observations (see Fig. 1 for references). Satellite-based estimates of large data sets provide longterm and near-global constraints for the Twomey effect and the $LWP$ adjustment. However, they are prone to numerous sources of uncertainties. These include, but are not limited to, uncertainties in $N_d$ changes for a given change in aerosol metric, the distortion of the true sensitivity due to relatively coarse retrieval scales (McComiskey and Feingold, 2012), and the covariability between meteorological factors and aerosol indices. Average forcing estimates for the Twomey effect alone range between

-0.2 to -1.0 W m$^{-2}$ (Quaas et al., 2008; Lebsock et al., 2008; Bellouin et al., 2013; McCoy et al., 2017). The $LWP$ adjustment may induce a partially compensating positive forcing to the Twomey effect, due to a decrease in cloud field $LWP$ (Gryspeerdt et al., 2019). Meanwhile, the LWP adjustment inside the convective cores of low clouds may be positive (Rosenfeld et al., 2019) which would locally amplify the aerosol-cloud forcing due to Twomey.

In the case of pollution tracks, the issue of covariability between confounding factors is avoided and a clear detection and

attribution of the cloud response to the aerosol perturbation itself, or at least to the corresponding change in $N_d$ is possible. Each individual track is associated with a spatially confined cloud response due to aerosol perturbations by ship or volcano plumes for a given set of meteorological conditions. However, these tracks are rare. It is estimated that merely 0.002 % of all ocean-going ships generate a ship track (Campmany et al., 2009). Though a recent estimate suggests that this number might underestimate the true ship track frequency (Yuan et al., 2019). Furthermore, they are only found within a narrow window of

meteorological conditions (Durkee et al., 2000). Therefore, while these estimates are prone to fewer uncertainties in detection and attribution of aerosol forcing, the representativeness of such estimates remains unclear.





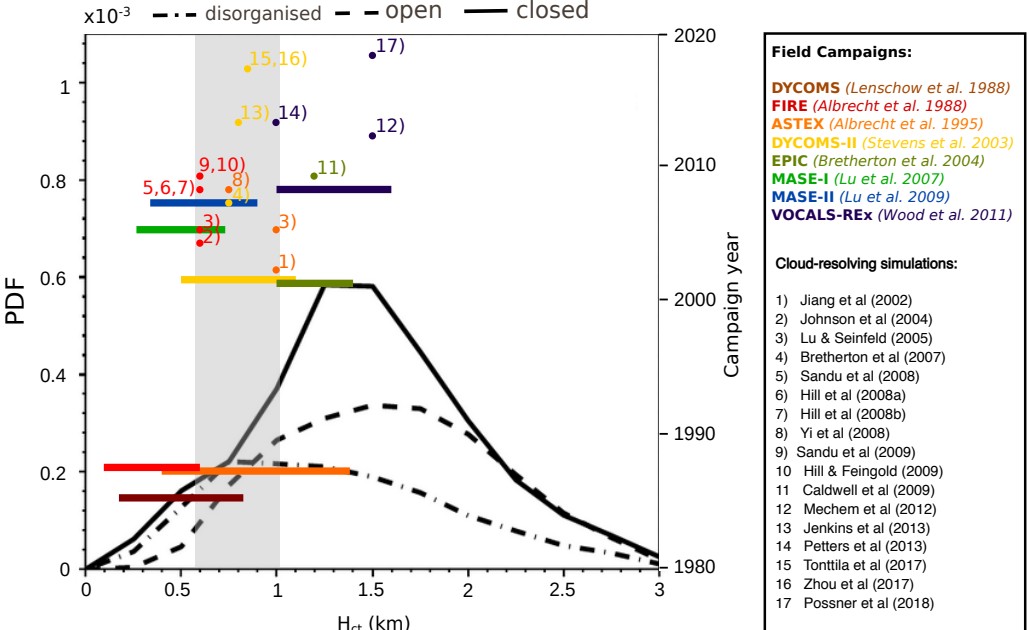

**Figure 1.** Probability density function for closed, open-cell and disorganised stratocumulus layers against cloud top height. This figure is adapted from Fig. 10 in Muhlbauer et al. (2014). Coloured bars denote range of cloud top heights sampled during each campaign listed in the legend. LES and cloud-resolving studies investigating aerosol-cloud-radiative interactions are colour-coded by the campaigns they are based on [with the exception of model study 8 which is based on an idealised profile]. Grey shading denotes narrow BL depth interval within which over 80 % of LES studies reside. Future analyses of past campaigns summarised in Zuidema et al. (2016) will likely increase the data points sampled in deeper BLs. References:Jiang et al. (2002); Johnson et al. (2004); Lu and Seinfeld (2005); Bretherton et al. (2007); Sandu et al. (2008); Hill et al. (2008); Hill and Dobbie (2008); Yi et al. (2008); Caldwell and Bretherton (2009); Sandu et al. (2009); Hill and Feingold (2009); Mechem et al. (2012); Jenkins et al. (2013)Petters et al. (2013); Tonttila et al. (2017); Zhou et al. (2017); Possner et al. (2018)Lenschow et al. (1988); Albrecht et al. (1988, 1995); Stevens et al. (2003); Bretherton et al. (2004); Lu et al. (2007, 2009); Wood et al. (2011)

The same holds true for estimates based on LES, cloud-resolving model studies, and field observations. At this resolution insights into the interplay between microphysical, radiative and thermodynamic processes can be obtained. Yet, the estimates

are representative for the conditions sampled and may not be valid generally, or at larger spatial scales. The LES community recently started to address these limitations, e.g. through extensive LES ensembles (Glassmeier et al., 2019). Here we would like to draw attention to the fact that previous analyses of LES and field campaigns have predominantly focused on shallow boundary layers. In Fig. 1 we show that most field campaigns and LES studies quantifying aerosol-cloud-radiative interactions have been conducted in BLs below 1 km in depth.

Fig. 1 shows the global distribution of stratocumulus regimes across BL depth (characterised here by cloud-top height), which





is comparable to the distribution of stratocumuli against BL depth in the subtropics alone (Fig. S1). In the subtropics merely 30% of stratocumuli reside at the depth range sampled in the field and studied within most LES. Results from merely two campaigns and even fewer LES studies are discussed within the literature that reside within a height range (deeper than 1 km) where over 70% of marine stratocumuli are found.

The lack of process studies in deep boundary layers, despite their prominence, motivates the question of exploring the dependence of cloud adjustments on BL depth. This is further supported by recent findings showing an explicit dependence of the $LWP$ adjustment on BL depth in pollution tracks (Toll et al., 2019). Here, we focus on regions dominated by marine stratocumuli, and explore these relationships within 10-year records in the subtropics. The data set is described in section 2. The change of mean cloud properties with BL depth is presented in section 3, while the impact of BL depth covariance with

$LWP$, and $N_d$ on the $LWP$ adjustment estimate is presented in section 4.

## 2 Data Description

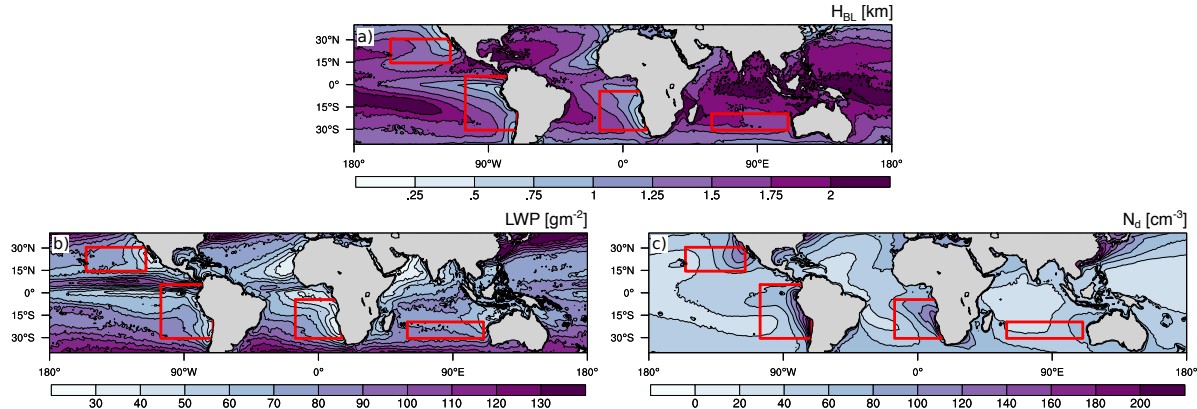

**Figure 2.** (a) Boundary layer height ($H_{BL}$), (b) liquid water path ($LWP$) and (c) cloud droplet number concentration ($N_d$) for the subtropical stratocumulus decks as defined in Eastman and Wood (2016) based on surface observations of Hahn and Warren (2007).

The relationship between $LWP$ and $N_d$ at different cloud depths is analysed in the semi-permanent stratocumulus regions of the subtropics (Fig. 2). The analysis is based on a 10-year climatology of daily in-cloud and radiation retrievals between 2007

and 2016, at a spatial resolution of $1 \times 1°$. Day-time in-cloud retrievals for LWP, $N_d$ and effective radius ($R_{eff}$) are obtained from the level 3 Moderate Resolution Imaging Spectroradiometer (MODIS) collection 6 product (King et al., 2003; Platnick et al., 2017). As in previous collections, independent retrievals of cloud optical depth and $R_{eff}$ are obtained using the visible and near-infrared radiances at 2.1 $\mu$m and 0.86 $\mu$m (Platnick et al., 2003).

The $R_{eff}$ retrieval is further used to distinguish between precipitating ($R_{eff} \geq 15\,\mu$m) and non-precipitating ($R_{eff} < 15\,\mu$m)

cloud scenes. For the year 2007 an independent retrieval of precipitation probability (Eastman et al., 2019) was available





(Fig. S2). During this year, the $R_{eff}$ criterion splits the data set into regimes where the precipitation probability remains below 50 % (equivalent to non-precipitating) and above 50 % (equivalent to precipitating).

$N_d$ is estimated based on the relationship established by Boers et al. (2006) and Bennartz (2007) for marine boundary layer clouds:


$$N_d = \sqrt{2} \frac{3}{4k\pi\rho_w} \Gamma_{eff}^{\frac{1}{2}} \frac{LWP^{\frac{1}{2}}}{R_{eff|top}^3} \tag{1}$$

where $\rho_w$ denotes the density of water, $\Gamma_{eff} = f_a d\Gamma_{ad}$ the effective rate of increase in adiabiatic liquid water content with increasing height and $R_{eff|top}$ denotes the effective radius at cloud top. All assumptions regarding the degree of adiabaticity and the proportionality constant $k$ between the true and effective $N_d$ are the same as in Eastman and Wood (2016).

The retrievals are restricted to sensor viewing angles between $0° - 65°$ (Grosvenor and Wood, 2014), which does not pose a strong constraint in the sub-tropics. The data is further limited to regions with high $CF$s exceeding 80 %. This restriction permits the best possible accuracy in $N_d$ retrievals, which assumes plane parallel clouds and restricts the analysis to large-scale stratocumulus cloud decks only, which have the largest radiative impact. All cloud properties are in-cloud mean values only, which are not weighted by areal $CF$ within each $1 \times 1°$ grid box.

The retrieval of BL height ($H_{BL}$) used in this study was first presented in Eastman and Wood (2016) and analysed in Eastman et al. (2017). The retrieval is based on a combination of MODIS and Cloud-Aerosol Lidar and Infrared Pathfinder Satellite Observations (CALIPSO) cloud retrievals (Vaughan et al., 2004). The Clouds and Earth's Radiant Energy System (CERES) Single Scanner Footprint One Degree (SSF1deg) retrievals of all-sky ($A_{toa}$) and clear-sky albedo ($A_{clr}$) based on the top-of-atmosphere shortwave fluxes (Kato et al., 2013) were used to estimate $A_{cld}$ from:


$$A_{toa} = CF * A_{cld} + (1 - CF) * A_{clr}. \tag{2}$$

It should be noted that the above equation can only provide an estimate of the cloud albedo. This definition, due to the separation of clear and cloudy skies, is highly sensitive to the definition of $CF$. $CF$ retrievals are afflicted with uncertainty, due to swelling of aerosols in the high relative humidity environment near cloud edges (Twohy et al., 2009), and rather than a 110 dichotomy, clear and cloud skies represent a continuum of albedo values (Charlson et al., 2007). Yet, Eq 3 has been shown to provide useful estimate of cloud albedo in the subtropical stratocumulus regions we focus on here.

Further environmental factors considered in this study, such as the lower tropospheric stability ($LTS$), and the free-troposphere relative humidity ($RH_{FT}$), are obtained from the Modern-Era Retrospective Analysis for Research and Applications Version 2 (MERRA2) reanalysis (Rienecker et al., 2011; Molod et al., 2015). The $LTS$ is defined as the change in potential temperature 115 between 700 hPa and the surface. Conditions are considered non-stable, if the change in potential temperature between these two pressure levels remains below 15 K. $RH_{FT}$ is diagnosed as the mean $RH$ between the inversion and 700 hPa. Environmental conditions are considered to be dry if $RH_{FT}$ falls below 50% and moist otherwise.



## 3   Covariance between Cloud Properties and Boundary Layer Depth



**Figure 3.** Scaling of (a) in-cloud albedo ($A_{cld}$), (b) liquid water path ($LWP$), (c) fraction of precipitating cloud scenes ($F_{prec}$), (d) effective radius ($R_{eff}$), and e) cloud droplet number concentration ($N_d$) against boundary layer depth ($H_{BL}$). Mean and standard deviation of the 10-year sub-tropical stratocumulus climatology (see Fig. 2) are shown in red within each height bin (100-m intervals). Fits across the climatology are superimposed in black (see table 1 for details on fitting parameters). The climatology of precipitating clouds is shown in blue.





Here, we analyse the climatological change in cloud properties of sub-tropical stratocumuli, such as $LWP$, $N_d$, $R_{eff}$, and $A_{cld}$, as a function of $H_{BL}$. Each cloud property is binned into $100\,\mathrm{m}$ $H_{BL}$ intervals within which the $10-\mathrm{year}$ mean and standard deviation are computed. The resulting climatology, which includes data from all four predominant stratocumulus cloud decks, is shown in Fig. 3. All cloud properties change significantly (at the 95 % level) with $H_{BL}$. The largest relative changes are observed for $LWP$ (Fig. 3b) and $N_d$ (Fig. 3e), while merely moderate and small changes are observed in $R_{eff}$ (Fig. 3d)

and $A_{cld}$ (Fig. 3a) respectively.

   The adiabatic $LWP$ ($LWP_{ad}$) scales with cloud depth ($H_c$) as $LWP_{ad} \propto H_c^2$, where $H_c = H_{BL} - H_b$ and $H_b$ denotes cloud base. Based on this relationship we regress $\ln LWP$ against $\ln H_{BL}$ to identify the exponent of the $LWP$–$H_{BL}$ relationship (Fig. 3b). Meanwhile, a simple linear regression is obtained for all other cloud climatologies (Fig. 3a,d,e). To understand the relative sensitivity amongst cloud properties to changes in $H_{BL}$, the slopes obtained by linear regression in the physical, as

opposed to logarithmic space, are scaled by the climatological mean (Table 1).

   As expected, larger $LWP$s are associated with deeper BLs (Fig. 3b). In particular we find climatological $LWP$ to scale as $LWP \propto H_{BL}^{0.33}$ (Table 1). Therefore $LWP$ scales considerably weaker with $H_{BL}$ than $H_c$. Combining these two relationships, it follows that in adiabatic clouds $H_c \propto H_{BL}^{0.17}$. That is, $H_c$ increases on average by merely $2\,\mathrm{m}$ for every $100\,\mathrm{m}$ increase in $H_{BL}$. Therefore $H_c$ seems largely independent of $H_{BL}$ variations.

Adiabaticity is known to change with cloud depth in marine stratocumuli (Merk et al., 2016; Braun et al., 2018). Furthermore, we find the likelihood of precipitation ($F_{prec}$) increases with $H_{BL}$ (Fig. 3c). Consequently, one might expect cloud adiabaticity to change as a function of $H_{BL}$ due to the change in $F_{prec}$. However, we find the $LWP - H_{BL}$ relationship is hardly impacted by precipitation (Table 1 columns 6 and 7). It also seems unlikely that the functional relationship between adiabaticity and $H_c$ would be sufficient to reduce the quadratic exponent of the $LWP$-$H_c$ relationship to that of the sub-linear exponent in the

$LWP$-$H_{BL}$ relationship. It therefore follows that $LWP$ scales very differently and seemingly independently with $H_{BL}$ and $H_c$ in marine sub-tropical stratocumuli.

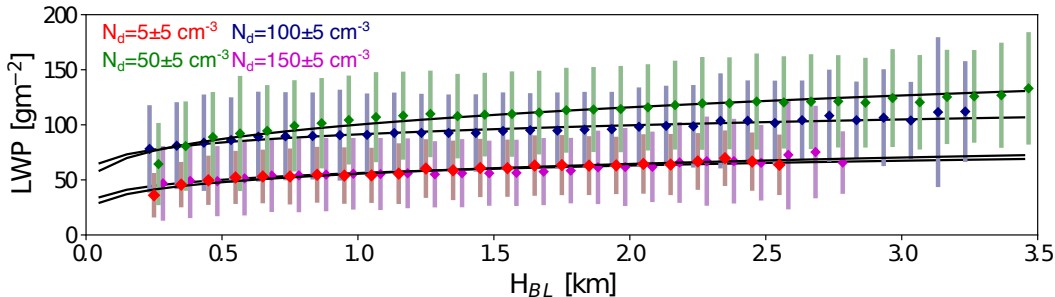

**Figure 4.** Liquid water path ($LWP$) against boundary layer height ($H_{BL}$). Markers denote climatological mean and bars denote the standard deviation. $LWP$ was binned in $H_{BL}$ for different cloud droplet number concentration ($N_d$) intervals. High and low $N_d$ collapse onto the same manifold while moderate $N_d$ manifolds are shifted towards higher $LWP$.





This has previously been formalised in the framework of *slow* and *fast manifolds*. The concept and evolution of manifolds was first applied to marine stratocumuli by Bretherton et al. (2010). Here, the cloud evolution is characterised by two distinct

timescales: the fast thermodynamic timescale ($< 1$ day), and the slow-varying timescale ($2-5$ days).

The thermodynamic timescale is representative for thermodynamic adjustments in $LWP$ and $H_c$ to a given set of conditions. On this timescale, Hc is predominantly constrained by the vertical displacement of $H_b$ rather than $H_{BL}$, as has been quantified by LWP budgets (Wood, 2007; van der Dussen et al., 2014; Ghonima et al., 2015; Hoffmann et al., submitted). $H_b$ in turn is governed by Clausius Clapeyron in response to fluctuations in BL humidity and temperature. This is consistent with the seem-

ingly weak relationship between $H_c$ and $H_{BL}$ inferred here from climatology. It further follows that the scaling relationship between $LWP$ and $H_c$ is representative for the thermodynamic adjustment occurring on short timescales.

Meanwhile, the multi-day evolution of the slow-manifold, which describes the adjustment timescale of $LWP$ and $H_{BL}$, characterises their evolution as a function of external drivers such as gradients in $SST$, $FT$ conditions, and large-scale advection. Bretherton et al. (2010) also found in LES experiments that $N_d$ could potentially impact the slow-manifold behaviour of marine

stratocumuli. This is in part observed within the 10-year climatology (Fig. 4). Although we do not observe a monotonic change in slow-manifolds as $N_d$ increases. Instead we find two distinct slow manifolds. One manifold for moderate $N_d$ ($N_d = 50-100\,\mathrm{cm}^{-3}$), and one for high ($N_d \geq 100\,\mathrm{cm}^{-3}$) as well as low $N_d$ ($N_d < 10\,\mathrm{cm}^{-3}$).

The LES experiments further showed that decoupling influenced the slow-manifold evolution. Decoupling is far more likely to occur in deep than shallow BLs (Bretherton and Wyant, 1997; Bretherton et al., 2010; Jones et al., 2011; Wood, 2012;

Dal Gesso et al., 2014). The cloud-top generated turbulent mixing is often insufficient in BLs deeper than $\sim 1$ km to extend the cloud-driven mixed layer towards the surface layer. In such BLs $H_b$ and $LCL$ may differ by $\mathcal{O}(10-100)$ m. Unfortunately we are unable to diagnose the state of decoupling within this data set. We can therefore not draw any conclusions on how decoupling impacts the climatological mean representation of the slow manifold.

$LWP$ is found to increase more rapidly with $H_{BL}$ under dry FT and non-stable lower troposphere conditions (Table 1 columns

$2-4$). This behaviour is consistent with cloud-scale observations (Eastman and Wood, 2018), simulations (Bretherton et al., 2013) and mixed-layer theory (Dal Gesso et al., 2014). Under low $RH_{FT}$ cloud-top cooling and cloud-top generated mixing are more effective. Therefore, a deeper and moister mixed layer associated with larger $LWP$ can be maintained. Thus, the reinforcement of the cloud through stronger radiative cooling has the stronger impact on $LWP$, than the increased drying through entrainment under low $RH_{FT}$ conditions. Similarly, the weaker buoyancy jump across the inversion under non-stable

lower troposphere conditions, likely induces less warming in the sub-cloud layer as the BL deepens, which corresponds to a weaker upward shift of the cloud base.

Meanwhile, deeper BLs are characterised by lower $N_d$ (Fig. 3e). As the BL deepens, $F_{prec}$ (Fig. 3c), and thus the $N_d$ sink through collision-coalescence processes, increases. Yet, $N_d$ primarily decreases with $H_{BL}$ in non-precipitating BLs. This suggests that precipitation scavenging is not the only constraint on $N_d$. In the absence of precipitation we attribute the anticor-

relation between $N_d$ and $H_{BL}$ to (i) the climatological deepening of the BL away from the cold upwelling zones near the coasts (Fig. 2a), and (ii) the increasing distance to continental sources of anthropogenic pollution, which manifest in a pronounced gradient in $N_d$ (Fig. 2c).





| Quantity | stability | | above cloud RH | | cloud-base precipitation | | all |
| --- | --- | --- | --- | --- | --- | --- | --- |
| | stable | non-stable | dry | moist | no-rain | rain | |
| $c_{\ln LWP}$ | 0.30 | 0.5 | 0.41 | 0.2 | 0.23 | 0.2 | 0.33 |
| $\overline{c_{R_{eff}}}$ | 0.10 | 0.08 | 0.13 | 0.13 | 0.07 | x | 0.13 |
| $\overline{c_{N_d}}$ | -0.2 | x | -0.3 | -0.28 | -0.13 | x | -0.3 |
| $\overline{c_{A_{cld}}}$ | 0.03 | 0.04 | 0.04 | x | 0.03 | 0.04 | 0.03 |
| $s_{lwp}$ (bivariate) | -0.31 | -0.19 | -0.30 | -0.15 | -0.28 | 0.14 | -0.28 |
| $s_{lwp}$ ($N_d$-only) | -0.33 | -0.22 | -0.37 | -0.16 | -0.28 | 0.28 | -0.33 |

**Table 1.** This table summarises the regime dependence. The relationship between cloud properties and $H_{BL}$ is determined logarithmically ($\ln \Psi \sim c\_\ln \Psi \times \ln H_{BL}$ for $\Psi = LWP$), or as normalised linear slopes ($\overline{c_\Psi} = c\_\Psi/\overline{\Psi}$ where $\Psi \sim c\_\Psi \times H_{BL}$ for $\Psi \in \{R_{eff}, N_d, A_{cld}\}$ and $\overline{\Psi}$ denotes the average). Slopes were determined by linear regression if and only if: (i) a significant fit was obtained at the 95-% confidence level and (ii) the fit explained at least 80% of the variance of the climatological relationship shown in Fig. 3. If no such fit is obtained, "x" is given. The regime dependence of $s_{lwp}$, which is defined and discussed in section 4, is summarised in the last two rows. Estimates for $s_{lwp}$ were either obtained by simple linear regression or by a bivariate fit taking the covariability between $LWP$, $N_d$ and $H_{BL}$ into account. All slopes are given to the significant digit which is determined based on the error of the respective fit.

This observed anticorrelation vanishes in a deregionalised and deseasonalised version of this data set (Fig. S3). Following Bender et al. (2016) we remove geographical and seasonal trends. In doing so, the significant anticorrelation between $N_d$ and $H_{BL}$ in non-precipitating clouds disappears. This further confirms, that the observed anticorrelation between $N_d$ and $H_{BL}$ in non-precipitating clouds is intrinsic to the climatology, but not a manifestation of BL or cloud adjustments.

The observed anticorrelation also disappears in the presence of precipitation (Fig. 3e and Table 1). The underlying causes of the negative relationship between $N_d$ and $H_{BL}$ climatologies are not impacted by precipitation. Yet the anticorrelation vanishes. This also holds for the deseasonalised and deregionalised $N_d$ climatology (Fig. S3). It follows that precipitation is the predominant constraint on climatological $N_d$ in sub-tropical marine stratocumuli at this scale. In addition $F_{prec}$ changes with $H_{BL}$. Thus, a significant negative slope manifests within the whole $N_d$ climatology ($\overline{c_{N_d}} = -0.3$), as the fraction of precipitating to non-precipitating clouds changes. Therefore, the $N_d$ climatology of all sub-tropical stratocumuli is constrained to first order by precipitation and to second order by the proximity to sources of cloud condensation nuclei.

The weakly positive scaling in $R_{eff}$ against $H_{BL}$ is consistent with the climatologies of $LWP$ and $N_d$. The decrease in $N_d$ is insufficient to offset the increase in $LWP$. The combined increase in $LWP$ and $R_{eff}$ with BL deepening results in a significant (Table 1), but inconsequential increase in $A_{cld}$ with $H_{BL}$. Stratocumuli with cloud tops above 1 km are associated with larger $LWP$, lower $N_d$, larger $R_{eff}$, and an elevated $A_{cld}$ of 0.01 as compared to stratocumuli with cloud tops below 1 km.





**Figure 5.** Climatology of $\ln LWP$ against $\ln N_d$ and $\ln H_{BL}$ for a) all sub-tropical stratocumuli (see Fig. 2), c) only precipitating stratocumuli and d) non-precipitating stratocumuli. Only points where the fraction of precipitating cloud scenes (panel b) exceeds 0.98 (or 0.02 respectively for non-precipitating clouds) within each $\ln N_d$–$\ln H_{BL}$ bin, are included in the climatology. At the top of each panel a,c,d) $\ln LWP$ binned in $\ln N_d$ is shown as in Gryspeerdt et al. (2019). A minimum number of 100 points within each $\ln N_d$–$\ln H_{BL}$ bin was required to be included in the climatology shown in opaque contours (a – d). The bivariate (simple linear) regression across the 2-dimensional (one-dimensional) climatology is shown in transparent contours (as black line) in panels a),c), and d). White and grey lines in panel a) denote the region of the phase space containing 80 % and 90 % of all data respectively. The slopes of all fits are summarised in Table 1.





## 4 Liquid Water Path Adjustment

Here we discuss the sensitivity of $LWP$ to changes in $N_d$ inferred from global climatology in the context of $H_{BL}$. In this discussion we make use of the commonly used susceptibility framework (Feingold et al., 2001) to characterise the susceptibility of $LWP$ as $s_{lwp} = \partial \ln LWP / \partial \ln N_d$.

Most stratocumuli within the sub-tropics (80 %) reside within an $LWP$ ($N_d$) range of $42 - 116 \, \mathrm{kg \, m^{-2}}$ ($20 - 164 \, \mathrm{cm^{-3}}$) at BL depths between $0.6 - 2.5 \, \mathrm{km}$. The largest climatological values of $LWP$ are found in deep BLs with low $N_d$ (Fig. 5a).

The increase in $LWP$ with $H_{BL}$ is consistent with climatology (Fig. 3b). The displayed sensitivity of $LWP$ to $N_d$ is potentially attributable to a multitude of competing factors; not all representative of cloud adjustments. The decrease in $LWP$ with increased levels of pollution has been noted multiple times in observations and various process hypotheses have been put forward.

Less polluted clouds could potentially be associated with weaker entrainment drying through the entrainment-sedimentation

feedback (Bretherton et al., 2007). Alternatively, increased rates of precipitation in cleaner environments could stabilise the cloud (Wood, 2012), which results in weaker overall cloud-top entrainment of dry sub-saturated air (Ackerman et al., 2004). Furthermore, recent results show that the strengthening of convective overturning in the sub-cloud layer through precipitation can also have a net positive impact on $LWP$ (Goren et al., 2018). All these, are examples of adjustments to the initial cloud microphysical response caused by an increase in droplet number. However, other factors not representative of cloud adjust-

ments, such as the climatological covariance between $H_{BL}$ and $N_d$ noted in Section 3, may impact $s_{lwp}$ estimates.

Here, we estimate $s_{lwp}$ by either fitting $\ln LWP$ against $\ln N_d$ as in (Gryspeerdt et al., 2019), or by fitting the two-dimensional surface of $\ln LWP$ against $\ln H_{BL}$ and $\ln N_d$. Both fits are simple linear, single- or bivariate regressions across the phase space containing 80 % of all data (Fig. 5a). Both fitting approaches yield similar and negative $s_{lwp}$ estimates. Taking the covariance with $H_{BL}$ into account merely reduces the magnitude of $s_{lwp}$ from $-0.33$, which is consistent with previous global estimates

of marine low-level clouds (Gryspeerdt et al., 2019), to $-0.28$ (Table 1). Therefore the bivariate fit of the entire climatology is likely subject to the same confounding factors impacting $LWP$ adjustments as in Gryspeerdt et al. (2019). Furthermore, the two predictor variables $H_{BL}$ and $N_d$ of the bivariate fit are not independent (Fig. 3e), which may bias the $s_{lwp}$ estimate. Especially in non-precipitating clouds the climatologies are negatively correlated.

Splitting the total phase space into precipitating and non-precipitating regimes clearly shows that $\ln LWP$ increases with $\ln N_d$

in precipitating clouds (Fig. 5c) and decreases in non-precipitating clouds (Fig. 5c). Furthermore, $s_{lwp}$ inferred from all clouds is dominated by the $LWP$ response in non-precipitating clouds. The opposing response in precipitating and non-precipitating regions is consistent with numerous previous studies (Albrecht, 1989; Ackerman et al., 2004; Bretherton et al., 2007; Wood, 2007; Wang et al., 2012; Suzuki et al., 2013; Gryspeerdt et al., 2019).





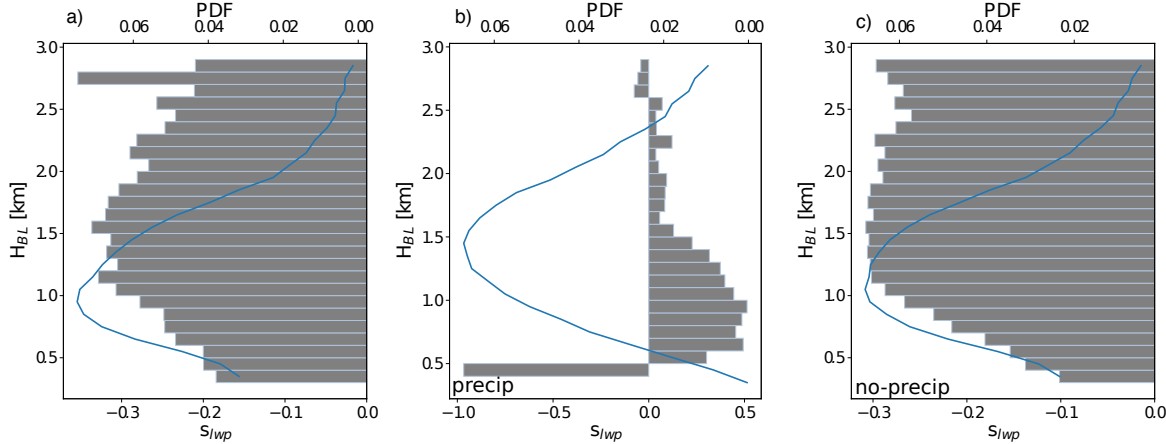

**Figure 6.** a) $s_{lwp}$ determined within $100$-m height intervals from whole sub-tropical stratocumuli climatology, b) precipitating clouds only, and c) non-precipitating clouds. $s_{lwp}$ was obtained with a bivariate linear regression of the $LWP$ surfaces shown in Fig. 5a), Fig. 5c) and Fig. 5d) within each height interval respectively. Only statistically significant results at the $95\%$ confidence level are shown. The probability density function (PDF) of the sub-tropical cloud climatology across the height bins is superimposed.

While $s_{lwp}$ in non-precipitating clouds was not sensitive to the fitting technique employed, the susceptibility to changes in $N_d$ in precipitating clouds halved by taking the covariance of $LWP$ with $H_{BL}$ into account (Table 1). In order to gain further insight into the potential variance of $s_{lwp}$ with $H_{BL}$, we obtained susceptibility estimates within constrained BL-depth ranges for precipitating clouds (6b) and non-precipitating clouds (Fig. 6c).

Analyses of ship tracks (Christensen and Stephens, 2011) and Lagrangian studies of cloud evolution (Eastman et al., 2017) have shown that $H_{BL}$ may increase under more polluted conditions. This, however, is not manifested within the $H_{BL}$-$N_d$ climatology (Fig. 3e). Particularly in non-precipitating clouds the climatological $H_{BL}$-$N_d$ relationship is not governed by processes representative of cloud adjustments (see section 3). Therefore, by constraining $s_{lwp}$ estimates in this manner, we attempt to remove some of the covariance between $LWP$, $N_d$ and $H_{BL}$ compounding the estimated strength of the $LWP$ adjustment. The $LWP$ adjustment in precipitating sub-tropical stratocumuli is considerably larger in BLs below $1.5\,\text{km}$ in depth than in deeper BLs (Fig. 6b). While $s_{lwp}$ may be as large as $0.48$, which constitutes a tremendous cloud adjustment in shallow BLs, it does not exceed $0.08$ in deep BL clouds. It should be noted that the large negative adjustment of $s_{lwp} = -1.0$ within the first height bin in Fig. 6b) is statistically significant, but characterises a very small sub-sample of the total climatology.

Meanwhile, $s_{lwp}$ increases in magnitude from $-0.1$ in BLs below $500\,\text{m}$ in altitude to $-0.34$ in BLs exceeding $1\,\text{km}$ in depth (Fig. 6a). Similarly, deep non-precipitating BLs exceeding $2\,\text{km}$ in depth are associated with less negative cloud adjustments than clouds situated within the climatologically dominant $H_{BL}$ depth range. These results are qualitatively consistent with the increase in $LWP$ susceptibility noted within pollution tracks around the globe (Toll et al., 2019). However, a quantitative comparison shows that $s_{lwp}$ estimates based on pollution tracks were considerably smaller than estimates inferred from cli-





matology. The $LWP$ adjustment increased from less than $-0.01 \pm 0.13$ in shallow BL clouds to $-0.1 \pm 0.13$ for a cloud top height of $2\,\text{km}$. Furthermore, the strengthening of $s_{lwp}$ was found to be continuous until much deeper cloud top heights up to

$3.5\,\text{km}$.

## 5   Conclusions

Isolating the $LWP$ adjustment due to changes in $N_d$ from potentially covarying meteorological factors has remained a significant hindrance in quantifying the radiative forcing of aerosol-cloud interactions. It also is a likely cause between diverging

estimates from low-cloud climatology, process-scale models, and pollution track estimates.

Here, we address whether $LWP$ adjustments vary with BL depth, and whether the susceptibility estimate is impacted by the covariance of cloud properties with $H_{BL}$. Like previous studies we find evidence for a positive change in $LWP$ with increasing $N_d$ in precipitating marine stratocumuli (Albrecht, 1989; Christensen and Stephens, 2011; Wang et al., 2012; Suzuki et al., 2013; Rosenfeld et al., 2019) which is consistent with the suppression of precipitation. Particularly in shallow precipitating BLs

($H_{BL} < 1\,\text{km}$) the estimated susceptibility can become very large ($s_{lwp} > 0.4$, Fig. 6). Such adjustments would correspond to a considerable enhancement of the negative cloud-radiative forcing. However, these shallow precipitating BLs are rare ($10\,\text{--}25\,\%$ of all cloud scenes analysed within the $10\,\text{--}$year climatology). Therefore, such cloud scenes are unlikely to govern the radiative forcing of aerosol-cloud interactions.

As was shown previously, the $LWP$ adjustment in marine sub-tropical stratocumuli is constrained by non-precipitating clouds.

Performing a bivariate fit of the $\ln LWP$ phase space, which removes any potential impact of the $LWP$-$H_{BL}$ covariance on estimates of $s_{lwp}$, was not found to provide substantially different results (Table 1) to previous global estimates of $s_{lwp}$ of marine low clouds (Michibata et al., 2016; Gryspeerdt et al., 2019). However, our analysis did show that $H_{BL}$ and $N_d$ climatologies are negatively correlated. Furthermore, this correlation is likely not generated by processes representative of cloud adjustments (section 3). At this point we cannot exclude that this covariance may impact estimates of $s_{lwp}$ inferred from climatology.

A further division of the entire phase space into BL-depth regimes showed that overall, cloud adjustments are less effective in shallow BLs. The potential increase in $LWP$ adjustment with BL depth has very recently been noted in pollution tracks (Toll et al., 2019). Here, we show that this behaviour may generalise to the whole climatology. Stratifying the $\ln LWP$ surface by BL depth, further closes the gap between $s_{lwp}$ estimates inferred from climatology and cloud-scale modelling. Shallow BLs, such as the ones sampled during ASTEX and DYCOMS-II (Fig. 1) are associated with $-0.22 < s_{lwp} < -0.1$ (Fig. 6a). This

is consistent with estimates of $s_{lwp}$ obtained in LES experiments of these campaigns (Ackerman et al., 2004; Bretherton et al., 2007).

Furthermore, we find that the change in $LWP$ with $H_{BL}$ is impacted by $N_d$ (Fig. 4). Changes in $LWP$ associated with changes in BL depth are characterised by a multi day timescale. This poses the question of whether the total change in $LWP$ to climatological changes in $N_d$ is fully characterised by rapid thermodynamic adjustments alone. We therefore suggest that

contrasting slow and fast manifold behaviour may aid the discussion to understand the diverging estimates of $s_{lwp}$ inferred



from climatology in observations (Gryspeerdt et al., 2019) and convection-resolving simulations (Sato et al., 2018), and $s_{lwp}$ inferred from short-lived anthropogenic pollution tracks (Toll et al., 2019).

In essence, we show that aerosol-cloud interactions may manifest differently in deep precipitating, and non-precipitating, marine BLs as compared to shallow BLs, and that rapid thermodynamic adjustments may not explain the entire response in $LWP$ to climatological changes in $N_d$. Furthermore, this work highlights the importance of understanding aerosol-cloud interactions in deep marine stratocumuli, which are underrepresented in currently analysed field data, numerical process models, and pollution tracks.

*Data availability.* MODIS cloud retrievals were obtained from: https://ladsweb.modaps.eosdis.nasa.gov/archive/allData/61/MYD08_D3 (last accessed January 2019). MERRA-2 data were downloaded from https://disc.gsfc.nasa.gov/datasets (last accessed March 2019). Albedo fields from CERES were downloaded from https://ceres.larc.nasa.gov/compare_products.php (last accessed November 2018)

*Author contributions.* AP conceived this study and wrote the manuscript. RE compiled the remote sensing retrievals for the study. AP and RE performed the analyses with input from FB and FG. All authors contributed to the discussion of the results, and editing of the manuscript.

*Competing interests.* The authors declare to have no competing interests.

*Acknowledgements.* We would like to thank D. McCoy for initial discussions of the project idea in the context of remote sensing products, L. Josipovic for his technical support in figure optimisation and F. Hoffmann for Mixed-layer theory discussions. A. Possner was funded by the Federal Ministry of Education and Research (BMBF) under the "Make our Planet Great Again – German Research Initiative", grant number 57429624, implemented by the German Academic Exchange Service (DAAD). R. Eastman would like to acknowledge the NASA grant NNXBAQ35G. Franziska Glassmeier acknowledges support by The Branco Weiss Fellowship – Society in Science, administered by the ETH Zurich, and by a Veni grant of the Dutch Research Council (NWO), and Frida Bender was supported by the Swedish Research Council, grant number 2018-04274.



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
