# Peer review of "Deconvolution of Boundary Layer Depth and Aerosol Constraints on Cloud Water Path in Subtropical Stratocumulus Decks"

_Atmospheric Chemistry and Physics, 2019_

## Referee Comment (RC1) · Anonymous Referee #1 · 21 Oct 2019

Comments on: Deconvolution of Boundary Layer Depth and Aerosol Constraints on Cloud Water Path in Subtropical Stratocumuli By Possner et al. In this paper the authors use 10 years of measurements (primarily from MODIS) to investigate the LWP response to changes in cloud droplet number concentrations and boundary layer depth in subtropical Sc. They show that, in agreement with previous studies, LWP increase (decrease) with Nd for precipitating (non-precipitating) clouds. The rate of decrease (or susceptibility) in LWP with Nd under non-precipitating conditions is shown to increase with the BL depth. The authors further claim that the deep BL conditions are underrepresented in previous studies, hence, previous estimations of LWP susceptibility may be underestimated. The paper is well written and presents important and timely results.

[Figure]

Hence, I support its publication after the following comment are addressed: General comment • One of the main conclusions/messages of this paper is that relatively deep BL clouds are underrepresented in studies of aerosol effect on LWP. However, there were previous LES studies simulating the transition between marine stratocumulus (Sc) to cumulus (Cu) and the aerosol effect on it. These studies include phases of deep BL. In addition, there were also many previous studies examining the aerosol effect on LWP in Cu clouds, with BL depth of 1.5 km and even more. I appreciate the focus on Sc, however, it looks to me as slightly artificial separation, especially if the focus is on relatively deep BL. I would expect that many of the physical processes acting in deep Sc and in Cu would be similar (as warm clouds cover the entire spectrum between Sc to Cu). For example, fig. 1 presents PDF of "disorganised" Sc. Looking on Fig. 1 of Muhlbauer et al., (2014), these disorganised Sc could definitely be (or at least be similar to) Cu. The fact that the data used here don't have any information on the decupling level in the boundary layer (L.163) only strength the relevancy of the Cu regime.

Specific comments • Abstract: I think it is better not to use "susceptibility" in the abstract without defining it as some readers may not know what it is. • L27: I think that decreased precipitation rates are a micro-physical effect and not "dynamic or thermodynamic adjustments". • Figure 1. The PDFs taken from Muhlbauer et al., (2014) are based on which data? • These processes (including the effect of the BL depth) were studied in Cu clouds.

Technical comments • L15: "due to be". • L16: "estimates in". • L168: "the stronger".

---

## Referee Comment (RC2) · Anonymous Referee #2 · 5 Nov 2019

Review of the manuscript numbered ACP-2019-833

Title: "Deconvolution of Boundary Layer Depth and Aerosol Constraints on Cloud Water Path in Subtropical Stratocumuli" written by Anna Possner et al. Manuscript number: "acp-2019-833". Decision: "Major revision"

In this study, the authors investigated the dependency of the Liquid Water Path (LWP) susceptibility of stratocumulus deck upon the boundary layer (BL) depth using 10 years (from 2007 to 2016) data. From their analyses, the authors elucidated that the susceptibility increases with deepening BL, and magnitude of susceptibility triples with deepening BL. LWP adjustment is one of the important topics in the climate science. And I

agree authors' suggestion that "the discussion based on the knowledge obtained from limited area of stratocumulus below shallow BL" can mislead the scientific community. So, I think this is an important study in the scientific community of the climate science. Most of the discussions in this manuscript is clear, and I agree most of the authors' suggestions. However, some discussions based on the previous process modeling study (slow- and fast- manifold mechanism) need to be modified. In addition, there are some technical problems. Based on the descriptions shown above, my decision is "not-so-major revision", and I encourage the authors to modify the manuscript. Detail comments are shown below.

General Comment:

1: The authors discuss the relationship between LWP-Hc and LWP-HBL in section 3, and try to interpret the difference between LWP-Hc and LWP-HBL relationship, and effects of aerosols on LWP-HBL relationship based on the slow and fast manifold mechanism. I agree that the discussions about slow and fast manifold mechanism are important for reducing the uncertainties of the cloud adjustment process. However, it is difficult for me to connect the results of this study to slow/fast manifold mechanism based on the analyses shown in the body of the manuscript. So, the discussion about the fast/slow manifold mechanism should be modified or removed from the manuscript.

2: The authors indicate the negative and positive LWP susceptibility in non-precipitating clouds and precipitating clouds, respectively (Table 1). This result supports the results of Gryspeerdt et al. (2019). In contrast, Chen et al. (2014), Michibata et al. (2016), Sato et al. (2018) indicated that the susceptibility is negative and positive or zero over precipitating and non-precipitating cloud areas. The authors should add some discussions about the reasons the inconsistency between the results of this study and the previous studies.

Specific Comment:

Title: This study targets on the stratocumulus "decks", and open cellar stratocumuli are

excluded from the analyses (CF > 80 %). So, I think the title with the word "deck" is better. For example, "Deconvolution of Boundary Layer Depth and Aerosol Constraints on Cloud Water Path in Subtropical Stratocumulus decks". This is just an example.

Line 63- 64: "In Fig. 1, we show that . . ." should be "Figure 1 shows that. . .". The Figure 1 is originated from Fig. 10 of Muhlbauer et al. (2014), not the authors' work.

Figure 1: LES intercomparison studies targeting on stratocumulus like Stevens et al. (2005); Ackerman et al. (2009), which are representative LES studies for DYCOMS and LES studies on VOLCALS case (Berner et al. 2013) should be added in the figures.

Line 68-69: "merely two campaigns and even fewer LES studies": Some concrete descriptions about the campaigns and LES studies targeting on the deep BL are helpful for readers to identify the previous studies targeting on deep BL.

Line 86: Fig. S2: The authors discuss Reff through the Fig. S2, but no data of Reff in Fig S2. Fig. S2 is same as Fig. S3, so, I think this is just a mistake. The authors should exchange the figure to correct one.

Figure 3: The data for non-precipitating case like Fig. S3 is useful for the reader, because the authors also discuss non-precipitating case in the body of the manuscript.

Line 140-157: As I mentioned in the general comment, it is difficult for me to connect the discussion of the LWP-Hc and LWP-HBL relationship to slow and fast manifold mechanism, through the results of this manuscript. I agree that the slow and fast manifold mechanism need to be considered when we discuss about the LWP adjustment. However, it is no evidence in this manuscript to justify that LWP-Hc and LWP-HBL relationship is regard as slow and fast manifold mechanism. The authors tried to justify through Hc-HBL relationship and Clausius Clapeyron, but I think these discussions could not convince readers that the LWP-Hc and LWP-HBL is regarded as the slow and fast manifold mechanism. In my understanding, the discussions about slow and fast manifold mechanism are not the main topic of the manuscript. So, the elimina-

tion of this part is one of the options. If the authors want to remain this part, I require the authors to add evidences to justify that LWP-Hc and LWP-HBL relationship can be regarded as slow and fast manifold mechanism.

Line 158-164: The authors discuss about the effect of the decoupling, but as the authors mentions in Line 162-163, no conclusion about the decoupling is obtained. So, I think this part is not necessary, and can be removed from the manuscript.

Table 1: Sample number for each column is helpful for readers. In addition, the regression statistics (e.g., error, residual, and so on) are helpful for readers. The information can be added as a supplemental material.

Line 174-177: In this part, the authors suggest that the anticorrelation between Nd and HBL is attributed to the climatological deepening of BL and increase of the distance to continental sources of anthropogenic pollution. However, there are no results to confirm these two suggestions. The trend of BL height and distance to continental sources are helpful for readers.

Line 178-180: The authors suggested that the anticorrelation between Nd and HBL vanishes in a deregionalised and deseasonalised version as shown in Fig. S3. However, weak anticorrelation, which is shown in black line of Fig. S3, is seen in climatological mean (red) and non-precipitating clouds (green) shown in Fig. S3. Is the word "anticorrelation" is same as "negative correlation"? If so, the authors should add some descriptions about the weak anticorrelation in climatological mean (red) and non-precipitating clouds (green) shown in Fig. S3. The value of slope for each case in Fig. S3 is helpful for readers. If not, please added the definition of anticorrelation more correctly.

Line 182-183: In this part, the authors suggest that the Nd and HBL climatology are not impacted by the precipitation, but the negative correlation in precipitating case is small but non-precipitating case is large. I think this means that the negative correlation is impacted.

Line 220-221: As I mentioned in the comment for Table 1, sample number for each column is helpful for readers.

Line 273-274: As I mentioned in general comment, it is difficult to regard LWP-Hc and LWP- HBL relationship as the fast and flow manifold mechanism from the results shown in the manuscript. Please do not misunderstanding, I agree the importance of slow and fast manifold mechanism.

Minor or technical Comment:

Figure 2: There are many contour lines around tropics, mid-latitude area, and ITCZ zone, and it is difficult to see the value over these areas. Of course, I understand that these areas are out of the scope of this study, but the figure need to be modified.

Line 147: "Hc" should be italic form and "c" should be subscript.

Figure 5: Unit of each variable in logarithmic is helpful for readers.

Line 153: Full spelling of SST (sea surface temperature) and FT (may by free troposphere) is helpful for readers.

Line 228: "(6b)" should be "(Fig. 6b)".

Line 234: I think the word "LWP adjustment" is used as slwp. Is this right? If so, slwp is easy to be understood.

Figures S1, S2, and S3: The label of Figure 1, 2 and 3 shown in supplemental material should be Figure S1, S2, and S3.

Reference:

Ackerman, A. S., and Coauthors, 2009: Large-Eddy Simulations of a Drizzling, Stratocumulus-Topped Marine Boundary Layer. Mon. Weather Rev., 137, 1083–1110, https://doi.org/10.1175/2008MWR2582.1. Berner, A. H., C. S. Bretherton, R. Wood, and A. Muhlbauer, 2013: Marine boundary layer cloud regimes and POC formation in a CRM coupled to a bulk aerosol scheme. Atmos. Chem. Phys., 13, 12549–12572, https://doi.org/10.5194/acp-13-12549-2013. Chen, Y.-C., M. W. Christensen, G. L. Stephens, and J. H. Seinfeld, 2014: Satellite-based estimate of global aerosol–cloud radiative forcing by marine warm clouds. Nat. Geosci., 7, 643–646, https://doi.org/10.1038/ngeo2214. Matsui, T., H. Masunaga, S. M. Kreidenweis, R. a. Pielke, W.-K. Tao, M. Chin, and Y. J. Kaufman, 2006: Satellite-based assessment of marine low cloud variability associated with aerosol, atmospheric stability, and the diurnal cycle. J. Geophys. Res., 111, D17204, https://doi.org/10.1029/2005JD006097. Michibata, T., K. Suzuki, Y. Sato, and T. Takemura, 2016: The source of discrepancies in aerosol–cloud–precipitation interactions between GCM and A-Train retrievals. Atmos. Chem. Phys., 16, 15413–15424, https://doi.org/10.5194/acp-16-15413-2016. Sato, Y., D. Goto, T. Michibata, K. Suzuki, T. Takemura, H. Tomita, and T. Nakajima, 2018: Aerosol effects on cloud water amounts were successfully simulated by a global cloud-system resolving model. Nat. Commun., 9, 985, https://doi.org/10.1038/s41467-018-03379-6. Stevens, B., and Coauthors, 2005: Evaluation of Large-Eddy Simulations via Observations of Nocturnal Marine Stratocumulus. Mon. Weather Rev., 133, 1443–1462, https://doi.org/10.1175/MWR2930.1.

---

## Author Comment (AC1) · 7 Feb 2020

Review Response for "Deconvolution of Boundary Layer Depth and Aerosol Constraints on Cloud Water Path in Subtropical Stratocumulus Decks"

We would like to thank both reviewers for their comments, which greatly helped to improve the clarity of this manuscript. Individual comments and concerns of each reviewer are addressed below. The colour code is as follows: reviewer comments and author response.

**Reviewer 1**:
Comments on: Deconvolution of Boundary Layer Depth and Aerosol Constraints on Cloud Water Path in Subtropical Stratocumuli By Possner et al. In this paper the authors use 10 years of measurements (primarily from MODIS) to investigate the LWP response to changes in cloud droplet number concentrations and boundary layer depth in subtropical Sc. They show that, in agreement with previous studies, LWP increase (decrease) with Nd for precipitating (non-precipitating) clouds. The rate of decrease (or susceptibility) in LWP with Nd under non-precipitating conditions is shown to increase with the BL depth. The authors further claim that the deep BL conditions are under-represented in previous studies, hence, previous estimations of LWP susceptibility may be underestimated. The paper is well written and presents important and timely results. Hence, I support its publication after the following comment are addressed:

We thank the reviewer for their comments, which we address individually below.

General comment: One of the main conclusions/messages of this paper is that relatively deep BL clouds are underrepresented in studies of aerosol effect on LWP. However, there were previous LES studies simulating the transition between marine stratocumulus (Sc) to cumulus (Cu) and the aerosol effect on it. These studies include phases of deep BL. In addition, there were also many previous studies examining the aerosol effect on LWP in Cu clouds, with BL depth of 1.5 km and even more. I appreciate the focus on Sc, however, it looks to me as slightly artificial separation, especially if the focus is on relatively deep BL. I would expect that many of the physical processes acting in deep Sc and in Cu would be similar (as warm clouds cover the entire spectrum between Sc to Cu). For example, fig. 1 presents PDF of "disorganised" Sc. Looking on Fig. 1 of Muhlbauer et al., (2014), these disorganised Sc could definitely be (or at least be similar to) Cu. The fact that the data used here don't have any information on the decoupling level in the boundary layer (L.163) only strength the relevancy of the Cu regime.

We would like to thank the reviewer for this comment as it touches on several aspects which were insufficiently addressed in the previously submitted version. Firstly, we entirely agree with the reviewer that in the case of shallow cumulus (Cu) fields detraining into thin cloud decks, such as can be found below a strong inversions, and precipitating stratocumulus (Sc), the distinction between the two regimes is obsolete. However, in case of Cu cloud fields characterised by a low cloud fraction (30-40%), the LWP adjustment may differ to that of Sc decks (CF>80%). Due to the increased fraction of sub-saturated clear-sky regions, the LWP adjustment seems governed by lateral entrainment and convergence processes (e.g. Jiang et al. (2006); Seifert et al. (2015)) as opposed to cloud thinning through vertical entrainment processes hypothesised to govern the LWP adjustment in marine stratocumuli. We agree that it is not clear at which point the distinction between Sc and Cu becomes somewhat semantic, but believe it is fair to contrast the behaviour of extensive Sc cloud decks of high cloud fraction to that of low-cloud fraction Cu. To address this concern in the manuscript, we have:

- removed the "disorganised" PDF from Fig. 1 (extension of Fig. 10 in Muehlbauer et al (2014)). This PDF is associated with cloud fields of an average cloud fraction of 40% and is thus likely to be more representative of shallow cumulus fields as opposed to the stratocumulus sheets studied here.
- Included results from the ATEX campaign, which is technically sampling detraining Cu, but there is no a priori reason to believe that the processes governing the LWP adjustment should be substantially different to that of Sc
- Changed the title of the manuscript to highlight the implications for Sc cloud decks to: "Deconvolution of Boundary Layer Depth and Aerosol Constraints on Cloud Water Path in Subtropical Stratocumulus Decks"
- included two paragraphs that summarise this discussion:

*"Fig. 1 shows the global distribution of stratocumulus regimes across BL depth which was characterised by Muhlbauer et al. (2014) in terms of cloud-top height (Fig. 10 in Muhlbauer et al. (2014)). The Muhlbauer et al. (2014) PDF is representative of all low-clouds over the oceans (see original paper for further methodology). We find the global PDF to be comparable to the distribution of stratocumuli against BL depth in the subtropics alone (Fig. S1). The PDF for disorganised clouds in Fig. 10 of Muhlbauer et al. (2014) was omitted here. These scenes were governed by broken cloud decks of low CF (CF = 40 %) resembling shallow cumuli rather than stratocumui.*
*The LW P adjustment within shallow cumuli seems governed by lateral entrainment effects and moisture gradients (e.g. Jiang et al. (2006); Seifert et al. (2015)). This is in stark contrast to stratocumulus cloud decks (CF 80 %) where the LW P adjustment is predominantly governed by vertical gradients in moisture, stability and aerosol. Thus, the LWP adjustment in shallow cumuli may differ from adjustments in stratocumuli, which is the focus of this study. The distinction between detraining shallow cumuli under strong inversions and precipitating stratocumuli becomes semantic in the case of cloud scenes associated with high cloud fraction. For this reason results of the Atlantic Trade Wind Experiment (ATEX) are included in Fig. 1."*

We further appreciate that several field campaigns and modelling studies have focused on Sc to Cu transitions and have explored the potential aerosol influence within these transitions. We have discussed this during the writing of this manuscript and decided to omit these studies for the following reasons:

(I) it is still debated to which extent precipitation and thus aerosol-sensitive cloud processes govern the breakup of the Sc deck into Cu cloud fields (e.g. Sandu & Stevens (2011) and Yamaguchi et al (2017)).

(ii) The LWP adjustment that originates from potentially delayed transitions likely only plays a secondary role to the inherent cloud fraction adjustment, which is not addressed in this study.

(iii) It is not clear how potential adjustments inferred from altered transition time scales relate to LWP adjustments within Sc decks generally.

References:
Sandu, I. and B. Stevens, 2011: On the Factors Modulating the Stratocumulus to Cumulus Transitions. *J. Atmos. Sci.,* **68**, 1865–1881, https://doi.org/10.1175/2011JAS3614.1.

Yamaguchi, T., Feingold, G., & Kazil, J. ( 2017). Stratocumulus to cumulus transition by drizzle. *Journal of Advances in Modeling Earth Systems*, 9, 2333– 2349, https://doi.org/10.1002/2017MS001104.

Specific comments:
Abstract: I think it is better not to use "susceptibility" in the abstract without defining it as some readers may not know what it is.

This section of the abstract was reworded as:

*" An unequivocal attribution of LW P adjustments to changes in aerosol concentration from climatology remains difficult due to the considerable covariance between meteorological conditions alongside changes in aerosol concentrations. We utilise the susceptibility framework to quantify the potential change in LW P adjustment with boundary layer (BL) depth in subtropical marine stratocumuli. We show that the LW P susceptibility, i.e. the relative change in LW P scaled by the relative change in cloud droplet number concentration,*
*in marine BLs triples in magnitude from −0.1 to −0.33 as the BL deepens."*

L27: I think that decreased precipitation rates are a micro-physical effect and not "dynamic or thermodynamic adjustments".

This has been rephrased as: "Through microphysical or thermodynamic adjustments [...]"

Figure 1. The PDFs taken from Muhlbauer et al., (2014) are based on which data? These processes (including the effect of the BL depth) were studied in Cu clouds.

Please see our response to the general comment above. Essentially we agree that these effects were studied in Cu clouds for different BL depths. Yet these regimes may not be identical in response to the Sc cloud decks studied here.

Technical comments:

L15: "due to be". Rephrased as "[…] due to changes […]".

L16: "estimates in". Rephrased  ("estimates off").

L168: "the stronger". Rephrased as "a larger".

Reviewer2:

In this study, the authors investigated the dependency of the Liquid Water Path (LWP) susceptibility of stratocumulus deck upon the boundary layer (BL) depth using 10 years (from 2007 to 2016) data. From their analyses, the authors elucidated that the susceptibility increases with deepening BL, and magnitude of susceptibility triples with deepening BL. LWP adjustment is one of the important topics in the climate science. And I agree authors' suggestion that "the discussion based on the knowledge obtained from limited area of stratocumulus below shallow BL" can mislead the scientific community. So, I think this is an important study in the scientific community of the climate science. Most of the discussions in this manuscript is clear, and I agree most of the authors' suggestions. However, some discussions based on the previous process modeling study (slow- and fast- manifold mechanism) need to be modified. In addition, there are some technical problems. Based on the descriptions shown above, my decision is "not-so-major revision", and I encourage the authors to modify the manuscript. Detail comments are shown below.

General Comment:
1: The authors discuss the relationship between LWP-Hc and LWP-HBL in section 3, and try to interpret the difference between LWP-Hc and LWP-HBL relationship, and effects of aerosols on LWP-HBL relationship based on the slow and fast manifold mechanism. I agree that the discussions about slow and fast manifold mechanism are important for reducing the uncertainties of the cloud adjustment process. However, it is difficult for me to connect the results of this study to slow/fast manifold mechanism based on the analyses shown in the body of the manuscript. So, the discussion about the fast/slow manifold mechanism should be modified or removed from the manuscript.

We appreciate the reviewers criticism. As this is not a key aspect and hard to prove from a Eulerian perspective, we decided to remove the fast/slow manifold discussion from the revised manuscript. We thus remove Fig. 4 of the submitted manuscript and revised the text accordingly. The revised text still addresses that changes in $H_c$ and $H_{BL}$ are constrained on different timescales and by different factors.

2: The authors indicate the negative and positive LWP susceptibility in non-precipitating clouds and precipitating clouds, respectively (Table 1). This result supports the results of Gryspeerdt et al. (2019). In contrast, Chen et al. (2014), Michibata et al. (2016), Sato et al. (2018) indicated that the susceptibility is negative and positive or zero over precipitating and non-precipitating cloud areas. The authors should add some discussions about the reasons the inconsistency between the results of this study and the previous studies.

We would like to thank the reviewer for raising this issue. To facilitate the discussion of this issue and also to contextualise our results of our revised Fig. 4 (previously Fig. 5 in original submission) we included an additional figure (Fig. 5) in the revised manuscript. Although we find predominantly -ve values of $s_{lwp}$, we also find +ve slopes in regions of moderate to high precipitation occurrence (which has been added to Fig. 2).
We agree that this is an important question to be resolved within the community, but that the resolution of this contradiction is beyond the scope of this study. We acknowledge and compare our results to the aforementioned studies. At this stage it would require a targeted effort by all participants to address to which degree the assumptions and methodologies differ and impact $s_{lwp}$ estimates.
In the revised manuscript we explicitly discuss this within a new paragraph added to the conclusions:
"Different remote-sensing-based estimates for slwp have been proposed. Their spatial distribution not only differs in magnitude, but also in sign among one another (e.g. Michibata et al. (2016) and Gryspeerdt et al. (2019)), as well as compared to Fig. 5 of this study. This is likely a result of different methodologies of categorising and processing different retrievals. Different methodologies to distinguish between precipitating and non-precipitating clouds, as well as different methods to retrieve and process Nd may impact slwp estimates. In particular, Nd remains a highly uncertain retrieval from space-born observations. For this study, we chose to limit the uncertainty of the physical retrieval of Nd while capturing as much of the variability in the subtropics as possible. Stricter filtering approaches may yield less retrieval uncertainty, but may imply a loss of some of the variability characteristic to the system. Either approach could influence slwp estimates. Thus our results, like previous studies, are subject to this uncertainty and remain to be verified in independent data sets."

Specific Comment:
Title: This study targets on the stratocumulus "decks", and open cellar stratocumuli are excluded from the analyses (CF > 80 %). So, I think the title with the word "deck" is better. For example, "Deconvolution of Boundary Layer Depth and Aerosol Constraints on Cloud Water Path in Subtropical Stratocumulus decks". This is just an example.

As suggested by the reviewer we changed the title to "Deconvolution of Boundary Layer Depth and Aerosol Constraints on Cloud Water Path in Subtropical Stratocumulus Decks".
It is worth noting though, that open cells may well be associated with cloud fractions of 80% and higher (see e.g. McCoy et al 2017).

Line 63- 64: "In Fig. 1, we show that . . ." should be "Figure 1 shows that. . .". The Figure 1 is originated from Fig. 10 of Muhlbauer et al. (2014), not the authors' work.

This was rephrased.

Figure 1: LES intercomparison studies targeting on stratocumulus like Stevens et al. (2005); Ackerman et al. (2009), which are representative LES studies for DYCOMS and LES studies on VOLCALS case (Berner et al. 2013) should be added in the figures.

We would like to thank the reviewer for the additional references. Ackerman et al (2009) and Berner et al (2013) were added to the figure. Although we agree that Stevens et al (2005) is one of *the* key publications in simulating stratocumuli, it does not explicitly focus on the impact of aerosols on cloud properties. This was a criterion for inclusion in this summary.
During this review we also added Xue et al (2008): "Aerosol Effects on Clouds, Precipitation, and the Organization of Shallow Cumulus Convection" and the ATEX campaign (Stevens et al 2001).

Line 68-69: "merely two campaigns and even fewer LES studies": Some concrete descriptions about the campaigns and LES studies targeting on the deep BL are helpful for readers to identify the previous studies targeting on deep BL.

We have expanded on this as follows:
"In the subtropics merely 30% of stratocumuli reside at the predominant depth range sampled in the field and studied within most LES. Results from merely three campaigns and few LES studies are discussed within the literature that reside within a height range deeper than 1 km where over 70% of marine stratocumuli are found. The campaigns containing measurements of deep stratocumulus cloud decks are ATEX, EPIC (East Pacific Investigation of Climate), and VOCALS-REx (VAMOS –Variability of the American Monsoons – Ocean-Cloud-Atmosphere-Land Study Regional Experiment). Merely 25% of all cloud-resolving modelling studies investigating the influence of aerosol concentrations on cloud properties in marine stratocumulus decks (i.e. Xue et al., 2008; Caldwell and Bretherton, 2009; Mechem et al., 2012; Berner et al., 2013; Possner et al., 2018) are based on deep BL field campaigns."

Line 86: Fig. S2: The authors discuss Reff through the Fig. S2, but no data of Reff in Fig S2. Fig. S2 is same as Fig. S3, so, I think this is just a mistake. The authors should exchange the figure to correct one.

We would like to thank the reviewer for catching this. The figure has been updated.

Figure 3: The data for non-precipitating case like Fig. S3 is useful for the reader, because the authors also discuss non-precipitating case in the body of the manuscript.

An additional figure was included in the supplementary material (Fig. S3) which shows the scaling relationships against BL depth for non-precipitating clouds in comparison to all clouds.

Line 140-157: As I mentioned in the general comment, it is difficult for me to connect the discussion of the LWP-Hc and LWP-HBL relationship to slow and fast manifold mechanism, through the results of this manuscript. I agree that the slow and fast manifold mechanism need to be considered when we discuss about the LWP adjustment. However, it is no evidence in this manuscript to justify that LWP-Hc and LWP-HBL relationship is regard as slow and fast manifold mechanism. The authors tried to justify through Hc-HBL relationship and Clausius Clapeyron, but I think these discussions could not convince readers that the LWP-Hc and LWP-HBL is regarded as the slow and fast manifold mechanism. In my understanding, the discussions about slow and fast manifold mechanism are not the main topic of the manuscript. So, the elimination of this part is one of the options. If the authors want to remain this part, I require

the authors to add evidences to justify that LWP-Hc and LWP-HBL relationship can be regarded as slow and fast manifold mechanism.

We have removed our discussion of slow and fast manifolds within the manuscript (see comment above).

Line 158-164: The authors discuss about the effect of the decoupling, but as the authors mentions in Line 162-163, no conclusion about the decoupling is obtained. So, I think this part is not necessary, and can be removed from the manuscript.

This section has been removed from the manuscript.

Table 1: Sample number for each column is helpful for readers. In addition, the regression statistics (e.g., error, residual, and so on) are helpful for readers. The information can be added as a supplemental material.
Following your suggestion additional regression statistics (sample number and statistical error) are put are included in a new table within the supplemental material Table S1.

Line 174-177: In this part, the authors suggest that the anticorrelation between Nd and HBL is attributed to the climatological deepening of BL and increase of the distance to continental sources of anthropogenic pollution. However, there are no results to confirm these two suggestions. The trend of BL height and distance to continental sources are helpful for readers.

Thank you for this suggestion. We now show this explicitly in a new figure in the Supplement (Fig. S4).

Line 178-180: The authors suggested that the anticorrelation between Nd and HBL vanishes in a deregionalised and deseasonalised version as shown in Fig. S3. However, weak anticorrelation, which is shown in black line of Fig. S3, is seen in climatological mean (red) and non-precipitating clouds (green) shown in Fig. S3. Is the word "anticorrelation" is same as "negative correlation"? If so, the authors should add some descriptions about the weak anticorrelation in climatological mean (red) and non-precipitating clouds (green) shown in Fig. S3. The value of slope for each case in Fig. S3 is helpful for readers. If not, please added the definition of anticorrelation more correctly.

We do not find a significant trend in the non-precipitating deregionalised/deseasonalised Nd-HBL relationship shown in Fig. S5 of the revised supplement. A significant slope is only determined for the full deregionalised/deseasonalised dataset (black line – fit to red points). However, this slope originates from the different weighting of the precipitating and non-precipitating curves as BL depth increases and the precipitating fraction of clouds increases. i.e. initially the all-cloud curve (red) is governed by non-precipitating clouds (green) in shallow BLs and increasingly influenced by precipitating clouds (blue) in deeper BLs. This is discussed in lines 203ff ("In addition...").

Line 182-183: In this part, the authors suggest that the Nd and HBL climatology are not impacted by the precipitation, but the negative correlation in precipitating case is small but non-precipitating case is large. I think this means that the negative correlation is impacted.

This section has been rephrased. The main message is that "[...], the $N_d$ climatology of all subtropical stratocumuli is constrained to first order by precipitation and to second order by the

proximity to sources of cloud condensation nuclei". We intended to state here that neither the BL deepening perpendicular to the coast (Fig. S4), nor the proximity to aerosol sources is directly impacted by precipitation. Yet the negative correlation between $N_d$ and $H_{BL}$ vanishes. Therefore, precipitation governs the $N_d$ signal despite the underlying processes which result in a negative correlation otherwise.

This section was rephrased for clarity as:

"The observed negative correlation also disappears in the presence of precipitation (Fig.3e and Table 1). Our two process hypotheses governing the negative correlation between $N_d$ and $H_{BL}$ are not impacted directly by precipitation. Yet the negative correlation vanishes. This also holds for the deseasonalised and deregionalised $N_d$ climatology (Fig. S4). It follows that precipitation is the predominant constraint on climatological $N_d$ in sub-tropical marine stratocumuli at this scale."

Line 220-221: As I mentioned in the comment for Table 1, sample number for each column is helpful for readers.
This information is now included.

Line 273-274: As I mentioned in general comment, it is difficult to regard LWP-Hc and LWP- HBL relationship as the fast and flow manifold mechanism from the results shown in the manuscript. Please do not misunderstanding, I agree the importance of slow and fast manifold mechanism.

Please see our response to your main comment.

Minor or technical Comment:
Figure 2: There are many contour lines around tropics, mid-latitude area, and ITCZ zone, and it is difficult to see the value over these areas. Of course, I understand that these areas are out of the scope of this study, but the figure need to be modified.
This figure has been revised for clarity.

Line 147: "Hc" should be italic form and "c" should be subscript.
Corrected.
Figure 5: Unit of each variable in logarithmic is helpful for readers.
Line 153: Full spelling of SST (sea surface temperature) and FT (may by free troposphere) is helpful for readers.
Was added.
Line 228: "(6b)" should be "(Fig. 6b)".
Corrected.
Line 234: I think the word "LWP adjustment" is used as slwp. Is this right? If so, slwp is easy to be understood.
Changed to slwp.
Figures S1, S2, and S3: The label of Figure 1, 2 and 3 shown in supplemental material should be Figure S1, S2, and S3.
Corrected.

Reference:
Ackerman, A. S., and Coauthors, 2009: Large-Eddy Simulations of a Drizzling, Stratocumulus-Topped Marine Boundary Layer. Mon. Weather Rev., 137, 1083–1110, https://doi.org/10.1175/2008MWR2582.1. Berner, A. H., C. S. Bretherton, R. Wood, and A. Muhlbauer, 2013: Marine boundary layer cloud regimes and POC formation in a CRM coupled to a bulk aerosol scheme. Atmos. Chem. Phys., 13, 12549–12572, https://doi.org/10.5194/acp-13-12549-2013.

Chen, Y.-C., M. W. Christensen, G. L. Stephens, and J. H. Seinfeld, 2014: Satellite-based estimate of global aerosol–cloud radiative forcing by marine warm clouds. Nat. Geosci., 7, 643–646, https://doi.org/10.1038/ngeo2214.

Matsui, T., H. Masunaga, S. M. Kreidenweis, R. a. Pielke, W.-K. Tao, M. Chin, and Y. J. Kaufman, 2006: Satellite-based assessment of marine low cloud variability associated with aerosol, atmospheric stability, and the diurnal cycle. J. Geophys. Res., 111, D17204, https://doi.org/10.1029/2005JD006097.

Michibata, T., K. Suzuki, Y. Sato, and T. Takemura, 2016: The source of discrepancies in aerosol–cloud–precipitation interactions between GCM and A-Train retrievals. Atmos. Chem. Phys., 16, 15413–15424, https://doi.org/10.5194/acp-16-15413-2016.

Sato, Y., D. Goto, T. Michibata, K. Suzuki, T. Takemura, H. Tomita, and T. Nakajima, 2018: Aerosol effects on cloud water amounts were successfully simulated by a global cloud-system resolving model. Nat. Commun., 9, 985, https://doi.org/10.1038/s41467-018-03379-6. Stevens, B., and Coauthors, 2005: Evaluation of Large-Eddy Simulations via Observations of Nocturnal Marine Stratocumulus. Mon. Weather Rev., 133, 1443–1462, https://doi.org/10.1175/MWR2930.1.

---

## Referee Report (RR1)

Review of the manuscript numbered ACP-2019-833

Title: "Deconvolution of Boundary Layer Depth and Aerosol Constraints on Cloud Water Path in Subtropical Stratocumulus Decks" written by Anna Possner et al.

Manuscript number: "acp-2019-833".

Decision: "Accept after very minor revision"

The authors have done a comprehensive job responding to my concerns and comments from another reviewer. The manuscript is much better than the previous version. So, I think that the manuscript can be accepted after some minor revisions shown below.

Minor or technical Comment:

Line 49: "effect" should be added after "Twomey".

Line 67: Muhlbauer et al. (2014)'s PDF

Line 127: Eq. 3 should be Eq. 2. (I cannot find Eq. 3)

Line 174: "(Fig. S3)" after the sentence "Yet, $N_d$ primarily decrease with HBL in non-precipitating BLs" is useful for readers.

Title of Section 5: I think "Conclusions and discussions" is better, because the authors discuss the difference between this study and previous studies.

---

## Author Response (AR2)

We would like to thank the reviewer for their comments below (shown in green). All edits were included within the final revision of the manuscript. In addition the following reference was included in the introduction as "in press":

Bellouin, N., Quaas, J., Gryspeerdt, E., Kinne, S., Stier, P., Watson-Parris, D., Boucher, O., Carslaw, K., Christensen, M., Daniau, A.-L.,
Dufresne, J.-L., Feingold, G., Fiedler, S., Forster, P., Gettelman, A., Haywood, J., Lohmann, U., Malavelle, F., Mauritsen, T., McCoy, D.,
Myhre, G., Mülmenstädt, J., Neubauer, D., Possner, A., Rugenstein, M., Sato, Y., Schulz, M., Schwartz, S., Sourdeval, O., Storelvmo,
T., Toll, V., Winker, D., and Stevens, B.: Bounding global aerosol radiative forcing of climate change, Reviews of Geophysics, in press, https://doi.org/10.1029/2019RG000660, 2019.

Review of the manuscript numbered ACP-2019-833

Title: "Deconvolution of Boundary Layer Depth and Aerosol Constraints on Cloud Water Path in Subtropical Stratocumulus Decks" written by Anna Possner et al.
Manuscript number: "acp-2019-833".
Decision: "Accept after very minor revision"

The authors have done a comprehensive job responding to my concerns and comments from another reviewer. The manuscript is much better than the previous version. So, I think that the manuscript can be accepted after some minor revisions shown below.
Minor or technical Comment:
Line 49: "effect" should be added after "Twomey".
Line 67: Muhlbauer et al. (2014)'s PDF
Line 127: Eq. 3 should be Eq. 2. (I cannot find Eq. 3)
Line 174: "(Fig. S3)" after the sentence "Yet, Nd primarily decrease with HBL in non-precipitating BLs" is useful for readers.
Title of Section 5: I think "Conclusions and discussions" is better, because the authors discuss the difference between this study and previous studies.